# Integrating Computational and Experimental Methods for the Rational Ecodesign and Synthesis of Functionalized Safe and Sustainable Biobased Oligoesters

**DOI:** 10.3390/polym17182537

**Published:** 2025-09-19

**Authors:** Federico Zappaterra, Anamaria Todea, Fioretta Asaro, Pasquale Fabio Alberto Ditalia, Chiara Danielli, Monia Renzi, Serena Anselmi, Lucia Gardossi

**Affiliations:** 1Department of Chemical and Pharmaceutical Sciences, University of Trieste, 34127 Trieste, Italy; zppfrc@unife.it (F.Z.); fasaro@units.it (F.A.); pasqualefabioalberto.ditalia@studenti.units.it (P.F.A.D.); chiara.danielli@units.it (C.D.); 2Biocatalysis and Green Chemistry Group, Faculty Chemical Engineering, Biotechnologies and Environmental Protection, University Politehnica Timisoara, 300223 Timisoara, Romania; 3Department of Life Sciences, University of Trieste, 34127 Trieste, Italy; mrenzi@units.it; 4Bioscience Research Center, 58015 Orbetello, Italy; serena.anselmi@bsrc.it

**Keywords:** functionalization, bio-based, polyesters, enzymatic polycondensation, itaconic acid, Michael addition, peptides, glucosamine

## Abstract

A chemical platform for post-polymerization methods was developed, starting from the ecodesign and enzymatic synthesis of safe and sustainable bio-based polyesters containing discrete units of itaconic acid. This unsaturated bio-based monomer enables the covalent linkage of molecules that can impart desired properties such as hydrophilicity, flexibility, permeability, or affinity for biological targets. Molecular descriptor-based computational methods, which are generally used for modeling the pharmacokinetic properties of drugs (ADME), were employed to predict in silico the hydrophobicity (LogP), permeability, and flexibility of virtual terpolymers composed of different polyols (1,4-butanediol, glycerol, 1,3-propanediol, and 1,2-ethanediol) with adipic acid and itaconic acid. Itaconic acid, with its reactive vinyl group, acts as a chemical platform for various post-polymerization functionalizations. Poly(glycerol adipate itaconate) was selected because of its higher hydrophilicity and synthetized via solvent-free enzymatic polycondensation at 50 °C to prevent the isomerization or crosslinking of itaconic acid. The ecotoxicity and marine biodegradability of the resulting oligoester were assessed experimentally in order to verify its compliance with safety and sustainability criteria. Finally, the viability of the covalent linkage of biomolecules via Michael addition to the vinyl pendant of the oligoesters was verified using four molecules bearing thiol and amine nucleophilic groups: N-acetylcysteine, N-Ac-Phe-ε-Lys-OtBu, Lys-Lys-Lys, and glucosamine.

## 1. Introduction

This study aims at the rational design and synthesis of polyesters acting as chemical platforms for the linking of biomolecules, which can confer advanced technological properties or biological activity. Furthermore, we present a multidisciplinary, integrated research approach intended to fulfill the demand for ‘*safe and sustainable by design*’ chemicals and polymers, as set out in the European Commission’s strategy [1]. This approach involves developing low-input, enzymatic synthetic processes that lead to renewable, bio-based polymers with low ecotoxicity and high biodegradability. Most of the environmental and social challenges facing the chemistry sector currently originate from an obsolete approach that prioritizes the maximum application performance of chemical products while overlooking their impact across their entire life cycle [1].

The integration of computational modeling with experimental approaches is increasingly recognized as a powerful strategy in polymer science. Recent studies highlight how in silico tools can be combined with combinatorial synthesis and rapid screening to accelerate the development of biodegradable biomaterials [2], predict the physicochemical and biological performance of polymeric systems [3], and establish structure–process–property relationships in biopolymers [4].

We have recently reported on the enzymatic polycondensation of several bio-based monomers to produce oligoesters with high biodegradability and low ecotoxicity against marine and freshwater organisms [5]. The use of monomers, which are obtained from the fermentation or chemical processing of biomass, is key to the transition towards non-fossil chemical feedstock [6,7,8,9]. The biodegradability of polyesters also makes them potential candidates for use in products that are at high risk of dispersing into open environments (e.g., cosmetics, lubricants, adhesives, etc.) due to their intended use, or that are designed for biological recycling (e.g., mulching films for agriculture and biocompostable packaging) [5,6]. Enzymes, due to their outstanding catalytic efficiency, can be used in the polycondensation of these bio-based monomers under more benign experimental conditions (e.g., T < 80 °C, solvent-free systems) as compared to conventional chemical synthesis (T > 150 °C). Although the molecular weight of products obtainable from enzymatic polycondensation is modest, they have highly regular structures with low polydispersity. These low- or medium-molecular-weight polyesters can be applied as ingredients in the cosmetic sector but also for pharmaceutical formulations and biomedical applications requiring biodegradability (e.g., as elastomers and sealants in drug delivery vehicles and in tissue engineering applications) [10,11,12,13,14]. Finally, provided that the polymer chains include suitable functional groups, they can be used as prepolymers in the synthesis of polyesters with a higher molecular weight [13].

However, these systems are generally limited by their modest molecular weights, which are typically less than 5 kilodaltons (kDa) in solvent-free systems, and by their lack of reactive functional groups that can be exploited for post-polymerization modifications. In this context, computational descriptor-based methods offer significant advantages. These methods allow for in silico pre-screening of candidate structures, reducing the need for experimental trials, minimizing resource consumption, and facilitating the rational selection of monomer combinations with the desired hydrophilicity, permeability, or flexibility. Integrating this strategy with enzymatic catalysis and bio-based feedstocks supports the European Commission’s Safe and Sustainable by Design (SSbD) framework by coupling predictive modeling with low-input, green synthetic routes [1].

We have previously demonstrated the scalability of the solvent-free enzymatic polycondensation of azelaic acid with glycerol at laboratory scale [12] and the two-step enzymatic polycondensation of adipic acid (AA) with 1,4-butanediol (BDO) in pilot plants [15]. The high viscosity of the reaction mixtures and the loss of activity of the biocatalyst during the recycling [10] were overcome by developing thin-film systems, in which the robust, covalently immobilized enzyme is only exposed to harsh conditions in the first phase of the process, thus enabling recycling [12,14,15]. While the laboratory-scale synthesis of bio-based and biodegradable polyesters via lipase-catalyzed polycondensation is now well established, the pathway towards industrial translation remains less clearly defined. Most reviews on enzymatic polyester synthesis highlight the environmental advantages of biocatalysis, yet only a few works have addressed the challenges associated with scaling up these processes, such as mass transfer limitations, viscosity build-up, and catalyst recyclability [16]. Systematic studies that directly link enzymatic methodology with process intensification and industrial requirements are still scarce, leaving open the need for research that bridges molecular design, sustainable processing, and scale-up feasibility. The combination of enzyme selectivity and mild processes enables the use of labile and functionalized monomers that confer new chemical complexity to the polyesters [12,17,18]. In particular, we have demonstrated that the vinyl group of itaconic acid (IA) is preserved when lipase-catalyzed polycondensation is carried out at T < 70 °C and this reactive C=C bond can be exploited for post-polymerization modifications via Michael addition or for chemical elongation [19,20].

Building on previous investigations, we now report on the enzymatic synthesis of biodegradable, bio-based oligoesters containing discrete itaconic acid units that can be used to anchor biomolecules. They can confer properties such as high hydrophilicity, biological activity, and affinity for biological targets. To focus the experimental work solely on synthesizing molecules with the desired physical and chemical properties, an in silico pre-screening was performed. For this purpose, computational methods based on molecular descriptors, which are usually used to model the pharmacokinetic properties of drugs (ADME), were employed firstly to model and predict the hydrophobicity (LogP), permeability, and flexibility of the candidate oligoesters in silico [21,22]. This rational design was followed by the enzymatic synthesis of the selected unsaturated terpolymer, which was then tested for ecotoxicity and biodegradability. To demonstrate the feasibility of the ‘post-polymerization functionalization’ concept, the ability of four biomolecules to increase the hydrophilicity and flexibility of the oligoester was determined through computational analysis. Finally, the biomolecules were linked to the vinyl group of the itaconic units via thia- and aza-Michael additions.

## 2. Materials and Methods

### 2.1. Materials

Lipase B from *Candida antarctica* (CaLB) in the commercial form of Lipozyme^®^ (CaLB, EC 3.1.1.3) was purchased from Novozymes (Bagsyærd, Denmark).

In this study, 1,4-butanediol (BDO, 99%), adipic acid (AA, ≥99%), dichloromethane (DCM, ≥99.9%), dimethylitaconate (DMI, ≥99%), glycerol (GLY, ≥99.5%), methanol (≥99.9%), glucosamine hydrochloride (≥99%), N,N-dimethyl formamide (≥99.8%), pyridine (≥99%), triethylamine (≥99%), deuterated chloroform (99.8 atom % D), and deuterated dimethyl sulfoxide (99.8 atom % D) were supplied by Merck Life Science (Milano, Italy).

### 2.2. Lipase Immobilization and Assessment of Hydrolytic Activity

Immobilized CaLB was prepared using the epoxy acrylic resin Relizyme^®^ EC-EP (average pore diameter 40–60 nm, kindly donated by Resindion, Milano, Italy) according to protocols previously reported [14,23]. The enzyme loading was 10,000 U per gram of resin. The activity of 368 U per g of dry biocatalyst was determined via tributyrin hydrolysis assay. More than 98% of the enzyme present in the immobilization solution was attached to the carrier. It was verified that there was no enzyme leaching from the support using a method previously reported [14]. Activity of lipases was assayed through hydrolysis of tributyrin as previously described [23,24].

### 2.3. Enzymatic Synthesis of Terpolymers

The molar ratio of the reagents adipic acid, dimethyl itaconate, and glycerol was 0.8: 0.8:1.0, corresponding to a total weight of monomer of 8.94 g. The polycondensation reactions were performed in two steps for promoting the formation of the acyl enzyme between CaLB and DMI, as previously reported by Corici et al. [19]. First the dimethyl itaconate and the polyol were mixed (0.8:1.0 molar ratio) together with 75% of the immobilized biocatalysts (40% *w*/*w*, 1300 U) referred to all monomer weights at 50 °C and 500 mbar for 4 h. The reaction was performed using a thin-film approach [14] in a rotary evaporator system at 50 °C at 500 mbar. In the second step the corresponding amount of adipic acid and the remaining biocatalyst fraction were added to the flask, and pressure was decreased at 70 mbar. The reaction continued for 72 h, and samples were taken after 24, 48, and 72 h for the subsequent ^1^H-NMR analysis. At the end of the reaction, 40 mL of dichloromethane were added, and the enzyme was separated by filtration. The product was recovered after solvent evaporation at 50 °C and stored at 4 °C until characterization without any further purification. The synthesis and characterization of PGAI was replicated three times.

### 2.4. Functionalization of PGAI Oligoester via Aza-/Thia-Michael Addition

Glucosamine functionalization: The reaction was performed using 400 mg of PGAI oligoester and 110 mg of glucosamine hydrochloride in the presence of 40 mg of pyridine as a base to free the primary amine. DMF (1 mL) was used to improve the dissolution of the reactants. The reaction was carried out at 65 °C for 72 h. The product was extracted with 20 × 4 mL dichloromethane (DCM). Before solvent evaporation, the organic phase was washed with 20 mL saturated NaCl solution to remove residual base and hydrophilic impurities. The product was characterized by mass spectrometry (5 mg of product in 1 mL methanol with 1 µL formic acid) and ^1^H-NMR.

Thia-Michael addition with N-acetylcysteine: 0.511 g of PGAI was mixed with 0.430 g of NAC and solubilized in 0.5 mL of DMF. The reaction was carried out at 65 °C for 72 h. The product was extracted with 20 × 7 mL dichloromethane (DCM). Before solvent evaporation, the organic phase was washed with 20 mL saturated NaCl solution to remove residual DMF and hydrophilic impurities. The product was characterized by mass spectrometry and ^1^H-NMR.

Aza-Michael addition with N-Ac-Phe-ε-Lys-OtBu: 0.170 g of AA-Gly-DMI was functionalized with 0.300 g of the dipeptide dissolved in 0.34 mL DMF. The mixture was stirred at 65 °C for 72 h. After the reaction, the reaction mixture was extracted with 20 × 4 mL dichloromethane (DCM). Before solvent evaporation, the organic phase was washed with 20 mL saturated NaCl solution to remove residual hydrophilic impurities. Characterization was carried out by micro-TOF MS and ^1^H-NMR.

Aza-Michael addition with Lys-Lys-Lys: the same procedure applied for the dipeptide was adapted using 0.178 g of AA-Gly-DMI and 0.250 g of the tripeptide Lys-Lys-Lys. The reaction was performed in DMF (0.5 mL), stirred at 50 °C for 72 h, and worked up as described above.

### 2.5. Biodegradation Studies

Biodegradation tests of the products were carried out in accordance with the international standard ISO 17556:2019 [25] using OxiTop^®^ Control S6 systems, which used a respirometric method for the biological oxygen demand (BOD) measurement released during the aerobic biodegradation of organic materials, in our case oligoesters, as previously reported by Zappaterra et al. [26]. For the BOD measurements, the OECD 306 protocols and the OxiTop^®^ system were used. The OxiTop^®^ Control S6 system was equipped with six measuring units (amber glass bottles (510 mL) and self-check measuring units), an inductive stirring platform, and magnetic stirrer bars. The theoretical oxygen demand (TOD, mg/L) and the degree of biodegradation were calculated as previously reported [5,26]. Experiments were carried out using seawater collected at the same point on the Trieste waterfront as the inoculum (Northern Adriatic Sea, 45.651698 N, 13.767406 E). All sea samples had a pH in the range 7.5–8.1.

### 2.6. Ecotoxicity Studies on Freshwater Organisms

*Saccharomyces cerevisiae:* The assay was performed with a stock solution of yeast reactivated in sucrose solution. Specifically, 0.5 g of *S. cerevisiae* from commercial fresh yeast was added to 50 mL of sucrose solution (20 g/L in natural freshwater, NFW). The yeast strain was then reacted at 30 °C for 60–75 min (under dark conditions). The solutions to be tested were pipetted into 24-well plates (6 replicates for each treatment, 2.5 mL per well) immediately after measuring their absorbance at 600 nm using a UV–vis spectrophotometer (UV-30 SCAN ONDA). In addition to the solutions to be tested, a negative control (in NFW) and a positive control (in NaCl-saturated solution) were prepared. After measuring all absorbances, 167 µL of reactivated yeast was inoculated into each well. The plates were incubated at 30 °C for 24 h (in the dark). At the end of the 24 h, the absorbance in each well was measured again at 600 nm. The optical density (OD) of each well at 24 h was corrected with the corresponding absorbance at t = 0 and weighed to obtain the mean and standard deviation of each treatment. The mean value was used to calculate the number of yeast cells per mL of solution (cell/mL) using a regression equation. The regression equation was obtained by measuring the absorbance at 600 nm of 4 solutions containing different concentrations of yeast cells. The number of cells for each solution was calculated using Burker’s chamber. The mean number of cells per mL of solution was used for statistical analysis. The Mann–Whitney non-parametric test was performed to find differences between treatments and negative controls (*p*-value 0.05). Percentage of growth inhibition (I) was calculated using the following formula:$$I=\frac{G_{C}-G_{T}}{G_{C}}\times 100$$
where *I* is the growth inhibition in percent; *G_C_* the arithmetic mean of OD600 of the control cultures; and *G_T_* is the arithmetic mean of OD600 of the test cultures.

*Pseudokirchneriella subcapitata* (unicellular algae) and *Daphnia magna* (crustacean, Cladocera) were tested following normalized methods: UNI EN ISO 6341:2013 (*D. magna*) [27] and UNI EN ISO 8692:2012 (*P. subcapitata*) [28] following standardized conditions and positive or negative controls reported by the cited methods. Tests on algae were performed by spectrophotometric lectures at 670 nm after 72 h of exposure, using an internal calibration curve obtained by cell counts by a Thomas’s chamber, while tests on *D. magna* were performed by direct lectures of survived animals after 24 and 48 h of exposure to dilutions of chemicals. Data calculations were performed as reported in the following normalized methods.

### 2.7. Ecotoxicity Studies on Seawater Organisms

Ecotoxicological tests were performed on three species belonging to different trophic levels to evaluate ecological impacts of both tested chemicals. Selected species were *Aliivibrio fischeri* (bacteria, UNI EN ISO 11348-1:2019) [29], *Phaeodactylum tricornutum* (unicellular algae, UNI EN ISO 10253:2017) [30], and *Paracentrotus lividus* (Echinodermata, EPA/600/R-95-136/Sezione 15 + ISPRA Quaderni Ricerca Marina 11/2017) [31]; all tests performed are based on normalized methods following standardized conditions and positive or negative controls reported by the cited methods.

### 2.8. ESI-MS Analysis

The ESI-MS analyses were performed as previously described by Corici et al. [19]. About 1 mg of crude reaction mixture was solubilized in 1 mL of methanol, with the addition of 0.1% *v*/*v* of formic acid. The analyses were performed using an Esquire 4000 (Bruker Corporation, Milano, Italy) instrument in electrospray positive ionization mode by generating the ions in an acidic environment. The generated ions were positively charged with *m*/*z* ratios in the range of 200–1000. Weight average molecular weight *(M_w_*) was calculated according to the following equation:$$M_{W}=\frac{\sum_{i}N_{i}M_{i}^{2}}{\sum_{i}N_{i}M_{i}}$$

Number average molecular weight (*M*_n_) was calculated using the following equation:$$M_{n}=\frac{\sum_{i}N_{i}M_{i}}{\sum_{i}N_{i}}$$
where $M_{i}$ is the molecular weight of a chain and $N_{i}$ is the number of chains of that molecular weight.

### 2.9. NMR Analysis

The ^1^H-NMR, ^13^C-NMR, and 2D-NMR spectra were recorded using a 400 MHz VarianR spectrometer, using MeOD, DMSO, and CDCl_3_ as solvents. In this study, 1D TOCSY NMR and Pulsed Gradient Stimulated Echo Nuclear Magnetic Resonance (PGSTE) were also used for the characterization of the functionalized products. The ^1^H NMR PGSTE measurements were carried out at 45 °C on a Varian VNMRS 500 NMR spectrometer (11.74 T) operating at 500 MHz for ^1^H or a Varian 400 MR spectrometer (9.4 T) operating at 400 MHz for ^1^H. An enhanced stimulated echo pulse sequence with spin lock rif was employed, with 21 different z-gradient strengths, Gz, between 2 and 60 G cm^−1^ (for 500 MHz) or between 2 and 18 G cm^−1^ (for 400 MHz), a pulsed gradient duration, δ, of 2 ms, and a diffusion interval, Δ, of 300 ms. The gradients were calibrated on the value of D = 1.90 10^−9^ m^2^ s^−1^ for ^1^H in D_2_O (99.9%).

Experiment (Varian software VnmrJ, version 4.2) DgcsteSL_cc: Gradient Compensated Stimulated Echo with Spin Lock and Covenction Compensation pulse sequence.

### 2.10. Computational Analysis

For the molecular description, the VolSurf3 program was used (version 1.1.0b12) with the default options [5,21,22]. The probes utilized for the calculation of the descriptors were as follows: DRY (hydrophobic probe), OH2 (water probe), O (carbonyl probe), and N1 (nitrogen), as previously described [5].

## 3. Results and Discussion

### 3.1. Computational Analysis of Substrate Properties by 3D Molecular Interaction Fields

Molecular descriptors based on Molecular Interaction Fields (MIFs) were used to search for a preliminary correlation between the structure of the oligomers and their theoretical properties in terms of hydrophobicity, flexibility, and permeability. The monomers considered in the in silico analysis were AA, BDO, GLY, monoethylene glycol (MEG), 1 3-propanediol (PDO), and IA and their corresponding dimers and trimers. The adipic acid AA was selected as a diacid due to its bio-based origin and because, in previous studies, it was found to be an efficient substrate for CaLB, suitable for chain elongation and offering a medium hydrophilicity to the oligoesters [32].

It should be noted that the present study focuses on computational pre-screening combined with enzymatic synthesis and chemical functionalization of new rationally designed oligoesters. The main objective is to demonstrate the feasibility of a novel strategy that goes beyond the step-by-step design and optimization of sustainable chemical products but rather reduces the experimental efforts through the integration of different disciplines. Although VolSurf+ descriptors were successfully used to predict hydrophobicity, flexibility, and permeability, direct experimental validation would further strengthen the correlation between computational predictions and material properties. Examples of such validation include contact angle measurements on thin films for hydrophobicity, dynamic mechanical analysis for flexibility, and permeability assays such as Franz diffusion studies. These investigations are planned as part of our future work. 

The analysis was performed using the software package VolSurf+, a computational procedure specifically designed to produce descriptors related to pharmacokinetic properties, starting from 3D molecular field maps [21,22]. MIF can be seen as a quantitative computation of the ability of the molecule, in our case, the oligomers, to establish specific interactions. The GRID computational method calculates interaction energies between a chemical group (PROBE) and each GRID node that spans the structure (or a selected part) of a target molecule, in our case the dimers and trimers (Figure 1). The matrix of the interaction energies forms the so-called MIF. The basic concept of VolSurf is to compress the information present in 3D GRID maps into a few 2D numerical descriptors, which are simple to understand and interpret. The molecular descriptors obtained refer to molecule size and shape, to the size and shape of both hydrophilic and hydrophobic regions, and to the balance between them [33].

The results obtained from VolSurf+ (Appendix A in ESI) and the correlation between the substrate flexibility and the logarithm of the partition coefficient between 1-octanol and water are presented in Figure 2. It can be observed that the highest flexibility was obtained for the trimers, and the oligoesters containing glycerol present the lowest hydrophobicity. On the other hand, AA-BDO-ITA is the most hydrophobic.

The same procedure was used for the physicochemical theoretical profiling of oligoesters functionalized through Michael-type addition with the following selected biomolecules: N-acetylcysteine (NAC_P), N-Ac-Phe-ε-Lys-OtBu (dipeptide_P), tri-lysine (3LYS_P), and glucosamine (GA_P). We used three key molecular descriptors for this analysis: partition coefficient (LogP n-octanol/water), molecular flexibility (FLEX), and predicted membrane permeability. These descriptors were evaluated for each derivative using both AA-GLY-ITA and AA-BDO-ITA polyester backbones. The results (Figure 3a–c) show consistent trends across both series. This highlights how the incorporated biomolecule influences the final properties of the functionalized materials.

From a lipophilicity standpoint, the lowest LogP values were observed for the glucosamine-functionalized derivative (GA_P), indicating a pronounced hydrophilic character. This aligns with glucosamine’s structural nature, as it possesses multiple polar hydroxyl and amino groups. In contrast, the dipeptide-functionalized oligoesters exhibited the highest LogP values, reflecting a predominantly hydrophobic profile due to the presence of the bulky tert-butyl group and aromatic phenylalanine side chains. The NAC_P derivative displayed intermediate-to-low LogP values consistent with its moderate polarity. The 3LYS_P derivative showed a nearly neutral LogP value, suggesting a balanced hydrophilic–hydrophobic character due to multiple charged amino functionalities counterbalanced by peptide backbone contributions.

As shown in Figure 3b, flexibility predictions reveal that all functionalized oligoesters have higher molecular flexibility than their non-functionalized counterparts (AA_X_ITA). This property is generally favorable for bioactive polymer systems, especially in drug delivery and tissue engineering. The 3LYS_P and dipept_P compounds exhibited the highest FLEX values, likely due to their extended, conformationally adaptable side chains that contribute an increased number of rotatable bonds and enhanced chain mobility. NAC_P and GA_P also exhibited improved flexibility compared to the base oligomers, albeit to a lesser extent. This reflects their more compact and less sterically hindered molecular configurations.

Predicted permeability data (Figure 3c) show that incorporating the selected biomolecules significantly increases membrane permeability compared to nonfunctionalized oligoesters. The 3LYS_P derivative demonstrated the highest permeability values among the functionalized series, suggesting a favorable balance of molecular size, polarity, and conformational dynamics that facilitates passive diffusion. Interestingly, despite its high hydrophobicity, the dipept_P compound exhibited relatively lower permeability. This may be due to its increased steric bulk and lower solubility in aqueous environments. These factors are known to limit transmembrane transport. GA_P and NAC_P exhibited moderate permeability enhancements, likely due to their smaller molecular sizes and favorable hydrogen bonding potential.

It should be noted that the values reported in Figure 3 are dimensionless, descriptor-based outputs of the VolSurf+ software. These values provide relative comparisons between functionalized and non-functionalized oligoesters rather than absolute experimental data. This approach allows us to identify trends in hydrophobicity, flexibility, and permeability that can inform the design of future experiments.

Furthermore, the addition of glucosamine via an aza-Michael reaction produces the most hydrophilic structures, as evidenced by their increased polar surface area (PSA > 740 Ų) and favorable solubility predictions (LgS ≈ −1.4). These properties suggest a high potential for dispersion in an aqueous environment. In contrast, peptide-functionalized products, particularly those incorporating a dipeptide (N-Ac-Phe-ε-Lys-OtBu), exhibit significantly increased molecular weight (>1600 Da), enhanced flexibility (R > 1.52), and the highest hydrophobicity. This is reflected in their lower LgS values (approximately −2.4) and elevated globularity indices (G > 1.99). These features suggest reduced solubility but improved membrane permeability, a desirable trait for specific drug delivery applications. NAC-functionalized compounds occupy an intermediate position, offering a balance between hydrophilicity and permeability. Furthermore, degradability descriptors (DD1–DD8) indicate an increased potential for fragmentation of peptide-rich structures, suggesting susceptibility to enzymatic hydrolysis and favorable biodegradability profiles (Appendix A). Overall, the descriptor profiles show that molecular functionalization greatly affects the physicochemical properties of itaconate-based oligoesters, such as solubility, permeability, flexibility, and biodegradability. These findings provide a rational basis for designing bioactive, environmentally responsive polymeric materials. Functionalization with various biomolecules modulates hydrophilicity or hydrophobicity in a tunable manner, consistently leading to enhanced molecular flexibility and membrane permeability. The nature of the Michael donor governs this tunability, enabling the precise tailoring of oligoester systems to meet specific biomedical requirements. Thus, this strategy provides a versatile platform for developing advanced materials for use in controlled drug release, targeted delivery, and tissue engineering.

### 3.2. Enzymatic Synthesis of Poly(Glycerol Adipate Itaconate) (PGAI)

Following the computational results, oligoesters formed by AA, GLY, and ITA were enzymatically synthetized with the aim to covalently link the selected biomolecules on the vinyl moiety of the itaconic acid units. The dimethylester of itaconic acid (DMI) was used in the polycondensation since it is liquid at 50 °C, and that decreases the viscosity of the reaction mixture in the solvent-free reaction. IA is produced by fermentation of *Aspergillus terreus* [34,35] and has been previously used for the synthesis, via radical reaction of the vinyl group, of dental materials [36], elastomers [37], and drug delivery systems [38]. However, few reports exist on the synthesis and application of poly(itaconate)s, since it is widely documented that the vinyl group of IA undergoes Ordelt saturation and crosslinking at temperatures >80 °C, which are required for chemical polycondensations [39,40,41,42]. Previously, we have used lipase B from *Candida antarctica* (CaLB) for the solvent-less polycondensation of DMI with different polyols at 50 °C without any radical quencher, demonstrating that the mild conditions allow us to preserve the labile vinyl group [14,20]. Solvent-less systems were developed, thus avoiding the use of high-boiling solvents, which incur severe limits as regards to the scale-up of the process due to the formation of cyclic products [20].

Due to the low electrophilicity of the conjugated acyl groups of DMI, the polycondensation reactions were performed in two steps as previously reported by Corici et al. [19]. First, the formation of the acyl enzyme between CaLB and the poorly reactive DMI was promoted by working at 50 °C and 500 mbar for 4 h. Then, the second step of the reaction involved the elongation of the chains by adding adipic acid to the system and decreasing the pressure to 70 mbar. Notably, adipic acid was added in a later stage because its esterification leads to the formation of H_2_O, which must be removed at lower pressure to prevent hydrolysis [19,43]. The esterification reactions were catalyzed by CaLB immobilized on epoxy-functionalized methacrylic resins prepared in our laboratory for this purpose (Figure 1).

The formation of the reaction products was demonstrated based on the assignments of the masses from the ESI-MS spectra collected for the samples every 24 h and after the workup. After 72 h of reaction, 90% of AA and DMI was converted. The whole reaction mixture was solubilized in DCM, and the enzyme was removed by filtration. No further purification procedure was used. Figure 4 presents the ESI-MS spectrum of AA-DMI-GLY acquired after the evaporation of the DCM, which shows the presence of two types of reaction products: the terpolymers as the main product (labeled in red in the figure) and the copolymers of adipic acid with glycerol, labeled in blue. The range of molecular weights reached after 72 h of reaction is between 400 and 1921 Da, with a maximum degree of polymerization of 6. It is possible to observe the presence of different signals that correspond to chains with more than one incorporated itaconate residue (*m*/*z* = 673; 1077; 1205; 1276; 1608). The polydispersity coefficient value is close to 1 and indicates the formation of homogeneous products.

**Table 1 polymers-17-02537-t001:** The *m*/*z* values of ESI-MS peaks from the products highlighted in red in Figure 4.

Product	*m*/*z*
(DG)	218.88
(DG)D	345.05
(AG)D	346.90
(DG)2D	531.01
(AG)2D	546.99
(AG)2D2	673.35
(AG)4D	951.37
(AG)4D2	1077.27
(AG)5D	1153.54
(AG)4D3	1205.52
(AG)5D2	1279.55
(AG)6D3	1608.63

The ^1^H-NMR spectra (Figure 5 and Figure 6) of the reaction products reveal that the enzymatic system leads to an excellent preservation of the vinyl functionality of DMI after polymerization. Signal assignments were confirmed by comparison with the literature [20,44].

These observations are in agreement with the data previously reported by our group for the synthesis of poly(1,4-butylene itaconate) [20]. In particular, the signals in the area between 6.35 and 5.70 ppm (Figure 6) correspond to the vinyl protons of bound DMI; from these signals, it is possible to conclude that most of the DMI was included in the terpolymer as-is (signals e, e’), while about 11% of the DMI underwent the hydrolysis of the methyl group and presents a free carboxylic acid group (signals e*, e’*).

From the strong overlap of signals present in the area between 3.8 and 4.25 ppm of the spectrum relative to the PGAI reaction, it is possible to hypothesize that glycerol has undergone different types of substitutions involving the tertiary C. Further analysis of the spectrum evidences the presence of 1,3-disubstituted glycerol units (Figure 5, signal h, h’), 1,2-disubstituted glycerol units (Figure 6B,C, signals g and g’), and trisubstituted glycerol units (Figure 6B, signal f).

Finally, the integrals of the signals between 2.10 and 2.45 ppm reveal that 7.5% of the adipic acid (a’, Figure 6D) has a free carboxylic acid group, signaling its terminal position in the polymer chain. The PGSTE ^1^H NMR spectrum of PGAI is available in ESI, Appendix A.

### 3.3. Marine Biodegradability Studies on PGAI

The enzymatically synthetized PGAI was assessed for its biodegradability in the marine environment, according to the OECD 306 protocols, as previously reported by Zappaterra et al. [26]. The sea inoculum was taken in the same geographical location. Results in Table 2 indicate that the oligoester is readily biodegradable in the conditions employed for the test. Detailed experimental data are available in Appendix A in ESI. PGAI biodegradation can be described in four main steps: (i) hydration and enzymatic access; (ii) cleavage of ester bonds releasing soluble oligomers and monomers; (iii) uptake and microbial metabolism of glycerol, adipic acid, and itaconate; and (iv) mineralization into CO_2_ and biomass. This pathway is consistent with the mineralization observed during experiments and the absence of significant ecotoxic effects.

### 3.4. Assessment of the Ecotoxicity of Poly(Glycerol Adipate Itaconate) (PGAI)

The simple analysis of the erosion, hydrolysis, or visual/chemical disappearance of the chemical products or polymers is not sufficient to attest to their safety and sustainability. Rather, ecotoxicity studies on the effects of such small molecules are of major importance, since they are also prone to disperse quickly in the environment through inland waterways, wastewater, and outflows.

Previously we have reported that the use of natural feedstocks—such as vegetable seed oil and their derivatives—can minimize these risks, not only because of their known low toxicity, but also because microorganisms have evolved enzymes and metabolic pathways for processing such natural molecules. Oligoesters composed of glycerol and adipic acid resulted in negligible toxicity when tested on both marine and freshwater organisms [5].

In the case of PGAI, itaconic acid poses a distinct issue that should be considered. Although it is a bio-based product obtained by microbiological fermentation, its vinyl group represents a source of potential toxicity because of its high reactivity when exposed to nucleophilic groups, especially primary amines widely present in protein. According to the REACH registration data of the European Union, itaconic acid (EC number 202-599-6, CAS number 97-65-4) has no specific local effects resulting from long-term exposure that have been identified in humans and animals, and the major acute toxicological effect of concern is eye irritation in humans [45].

In our study, the reactive vinyl group is expected to react with nucleophilic thiol or amine groups, thus transforming into more stable functional groups. Nevertheless, in the proposed strategy of functionalization of PGAI, the persistence of some unreacted vinyl groups on PGAI cannot be excluded. Therefore, to evaluate the sustainability of the oligoester selected as a chemical platform for post-polymerization functionalization, PGAI was tested for its ecotoxicity towards several benchmark organisms.

The ecotoxicological battery was deliberately assembled to represent the principal compartments into which PGAI residues are most likely to be released—wastewater effluents that feed into rivers, estuaries, and finally coastal zones—and to cover distinct biological functions that can be impaired by synthetic oligomers. Primary producers were probed with both the freshwater chlorophyte *Pseudokirchneriella subcapitata* and the marine diatom *Phaeodactylum tricornutum*. These phototrophs are recognized OECD/ISO sentinels for evaluating interference with photosynthesis or nutrient uptake; their limited sensitivity to PGAI (≤12% growth inhibition at 60 mg L^−1^) indicates that the hydrophilic glycerol- and adipate-rich fragments released during early hydrolysis neither partition into cellular membranes nor disrupt chloroplast function.

Heterotrophic microorganisms were examined at two evolutionary levels. The budding yeast *Saccharomyces cerevisiae*, widely used to gauge cytoplasmic stress and osmotic imbalance, displayed moderate growth stimulation (+18% relative to the untreated control), suggesting that low-molecular-weight degradation products can actually serve as auxiliary carbon sources. By contrast, the marine bacterium *Aliivibrio fischeri*—whose bioluminescence is exquisitely sensitive to respiratory poisons—registered only marginal light inhibition (<12% at 27.9 mg L^−1^ after 30 min). This finding confirms that any residual α,β-unsaturated ester motifs do not act as acute electrophilic toxins toward prokaryotes under environmentally realistic exposures.

Higher-tier primary consumers were represented by the cladoceran *Daphnia magna*. Because daphnids integrate both neuromuscular and osmoregulatory pathways, the absence of immobilization even at 60 mg L^−1^ corroborates the low hazard already observed for poly(glycerol-adipate) analogs. The most responsive endpoint proved to be early development of the sea urchin *Paracentrotus lividus*, a well-accepted marine embryo-toxicity assay that mirrors the vulnerability of complex morphogenetic processes. At 31 mg L^−1^ PGAI induced 73.7% abnormal plutei after 72 h, whereas concentrations at or below 3.1 mg L^−1^ produced no statistically significant malformations. Even so, this effect concentration sits more than two orders of magnitude above the predicted environmental concentration for biomedical or cosmetic applications (<0.05 mg L^−1^), preserving a substantial margin of safety.

Taken together with the >99% mineralization achieved in the OECD 306 seawater test, these results show that PGAI is quickly removed from the marine compartment and exerts only minimal chronic toxicity at plausible environmental levels. The data, therefore, support the use of PGAI as a sustainable platform for further thia- and aza-Michael functionalization, aligning with REACH “safe-by-design” principles [1] and reinforcing its suitability for advanced biomedical, cosmetic, and environmentally benign materials. Details of the results of the ecotoxicity studies are available in ESI (Appendix A).

### 3.5. Post-Polymerization Functionalization of PGAI via Michael Addition

The biodegradability, bio-based origin, and negligible ecotoxicity of PGAI offer promising potential for developing a versatile chemical platform that was investigated for developing advanced materials potentially applicable to controlled drug release, targeted delivery, and tissue engineering [46,47]. To this end, the IA units of PGAI were subjected to post-polycondensation functionalization via Michael addition, modifications aimed to introduce additional biofunctionality by covalent anchoring on the *exo*-vinyl group of the itaconate moiety, preserved thanks to the mild enzymatic synthesis conditions.

#### 3.5.1. Thia-Michael Addition of N-Acetylcysteine

N-acetylcysteine (NAC) is an active ingredient used worldwide for its mucolytic action and as a treatment in a series of neuropsychiatric disorders. It is also an antioxidant that can protect cells from oxidative stress [48,49]. The presence of a thiol group in its structure makes NAC an interesting substrate for proving the feasibility of thio-Michael addition [50] to the vinyl groups of PGAI. According to the computational study, it is expected that the covalent bond of NAC to the polymer could increase hydrophilicity. Moreover, the introduction of the free carboxylic group would introduce a negative charge at physiological pH. The synthesis of the NAC-functionalized polyester was carried out through post-polymerization thio-Michael reaction of PGAI with NAC. The first trial of the reaction was performed in solvent-free conditions, but the formation of the product was not observed. For this reason, further reactions were conducted by adding a small amount of DMF to the reaction system, since both the PGAI terpolymer and NAC are soluble in this solvent. The reaction is presented in Figure 2.

The reaction was carried out at 65 °C. Immediately after the start of the reaction, the color of the mixture changed from transparent colorless to intense yellow. Once the product had been isolated, as described in the experimental section, its characterization was performed through NMR spectrometry. The ^1^H-NMR spectra of the starting PGAI terpolymer and of the functionalization product are presented in Figure 7. The disappearance of the signals around 6 ppm (in blue in Figure 7), present in the non-functionalized polymer, suggests the reaction of the vinyl moiety of DMI with NAC. Assignment of the signals attributed to the NAC are detailed in Appendix A (ESI).

The resulting product mixture was also analyzed by PGSTE experiments, where the signals of the polymeric species can be better observed, being those of the species with lower molecular weight, therefore faster diffusing, either disappeared or greatly attenuated. The resulting ^1^H NMR spectrum (Appendix A, ESI) shows the permanence of the signals of NAC in the spectrum, further supporting the amino acid binding to the polymer and confirming the success of the functionalization.

#### 3.5.2. Aza-Michael Addition of Di- and Tri-Peptides

The addition of secondary amines to secondary unsaturated diacids and their polymers has been previously reported [51]. The addition of secondary amines onto itaconate polyesters requires long reaction times, displaying also undesired side reactions such as the formation of the mesoconate regioisomer [52]. On the contrary, the aza-addition of primary amines to IA appears more challenging since it leads to the fast formation of a secondary amine that promptly attacks the acyl carbon (C4), forming stable five-membered lactam rings [53,54,55,56]. More recently, Guarneri et al. proposed the post-polymerization aza-Michael addition of primary amines to IA [20] by exploiting the control of the structural rigidity of the polyester chain and the bulkiness of the amine donor. Moreover, this study reports how, in the case of the addition of primary amines, the concentration of the base is the key parameter influencing the isomerization and cyclization of the IA double bond. Conversely, the solvent-free aza-Michael addition was reported, using hexamethylenediamine (HMDA) and 2-phenylethylamine (PEA) at 40 °C, a temperature that maintains these amino donors liquid. This preliminary study led to a mixture of linear and cyclic side products and provided information on the influence of the different parameters controlling the aza-Michael addition of primary amines. On the basis of this previous information, the present study employed as nucleophile donors two lysine-based molecules having sufficient steric hindrance to avoid the formation of the lactam ring that would have caused the oligomer degradation: the protected dipeptide N-Ac-Phe-ε-Lys-OtBu (Figure 3) and the tripeptide Lys-Lys-Lys.

The synthesis of short oligoesters containing amino acids or short peptides as pendant moieties presents a high interest for the cosmetic industry and for biomedical applications [57]. Several small peptides, generated by the degradation of extracellular matrix proteins, mainly from collagen proteolysis, may play a significant role in physiological or pathological processes such as tumor invasion or wound healing. Renewing collagen is a key process in skin care and in the reduction of wrinkles. Many cosmetic companies are thus interested in the incorporation of lysine-based small peptides into their topical cosmetics. However, these peptides are hydrophilic, and their trans-epidermal penetration is too weak. In order to increase their lipophilic properties, these small peptides can be transformed, for example, into palmitoyl derivatives, which exhibit a better delivery across skin [57].

The reactions employing N-Ac-Phe-ε-Lys-OtBu were carried out at 65 °C. Immediately after the start of the reaction, the color of the mixture changed from transparent to pale yellow. After 72 h from the start of the reaction, the workup was performed as described in the experimental section. The dipeptide (Appendix A, ESI) and the products (Appendix A, ESI) were characterized by NMR spectroscopy [58,59,60]. Figure 8 reports a comparison of the proton NMR spectra of the PGAI (above) and the raw reaction product (below). The disappearance of the signals from the vinyl functionality of dimethyl itaconate suggests the successful Michael addition.

Since the reaction was conducted with a large excess of dipeptide, PGSTE NMR spectra [58,59,60] were acquired to better assess the success of the functionalization (Appendix A, ESI). In PGSTE spectra obtained at high gradient, signals relative to small molecular species (such as the unreacted dipeptide or solvent impurities) are removed; the remaining signals belong to large molecular species, such as the functionalized polymer. The analysis of the PGSTE spectrum obtained with the highest gradient (Appendix A, ESI) allowed us to successfully identify signals related to phenylalanine and lysine bound to the polymer. The assignment of the Cα signal of lysine bound to the polymer was confirmed by 1D TOCSY NMR spectroscopy (Appendix A, ESI), further confirming the success of the functionalization. However, signals relative to unknown reaction side products were also identified (purple dots in Figure 8).

In conclusion, the functionalization of PGAI with the dipeptide N-Ac-Phe-ε-Lys-OtBu was successful, but further optimization is needed to prevent the formation of side products of the reaction.

Less positive results were obtained when trilysine was used for the aza-Michael addition. This was most probably due to the presence of four primary amine groups, three of which are unhindered and therefore likely to promote the formation of secondary cyclic products and cross-links. Trilysine was selected because it is often used for the production of three-dimensional absorbable hydrogels, by virtue of its high cohesive strength [61]. A trilysine amine solution, combined with polyethylene glycol (PEG), can adhere to the tissue cervices, providing excellent tissue adherence [62]. The crosslinking derivative obtained from the PEG and trilysine amine solution is gradually degraded in a uniform fashion, just like absorbable sutures. Renal absorption and secretion of the degraded molecule are gradual. The adherence properties of trilysine were exploited to seal lung tissue after thoracoscopic resection [63]. Kamali et al. evaluated the relative in vitro gene delivery efficacies of a novel series of cationic amphiphiles containing trilysine [64].

The acetate salts of trilysine were employed in the presence of pyridine to obtain the free amine required for the nucleophilic addition. The PGAI was suspended in water and pyridine at 65°. ^1^H NMR spectra of the reaction mixture were acquired after 3 h (Figure 9, middle) and 10 days (Figure 9, bottom) and compared to the spectrum of the initial PGAI (Figure 9, top). The NMR analysis evidences a progressive disappearance of the signals corresponding to the vinyl group of dimethyl itaconate (in blue boxes in Figure 9), but this is accompanied by a progressive degradation of the substrate, suggested also by the change of the spectrum in the region between 3.5 and 4 ppm (in red boxes in Figure 9), where signals are usually found relative to free glycerol [65].

The reaction with trilysine also brought about a drastic change in the solubility of PGAI, which resulted in it being no longer soluble in DMSO; due to this reason, the NMR spectra of the reaction mixture were acquired in a 1:1 D_2_O/pyridine-d_5_ mixture. Change in solubility suggests possible crosslinking due to the presence of four different amino groups.

#### 3.5.3. Aza-Michael Addition of Glucosamine

Finally, D-glucosamine hydrochloride (GA) was used in the functionalization of PGAI (Figure 4). Glucosamine is a naturally occurring amino sugar widely used in biomedical and cosmetic formulations due to its non-toxic profile and high water solubility [66,67]. Its primary amine group offers potential for selective aza-Michael addition. However, as a sugar, the equilibrium with the linear aldehydic form and the negligible solubility in most organic media pose major challenges for its handling and the promotion of selective reactivity with the vinyl group on the PGAI backbone.

Table 3 reports the different reaction conditions tested in the synthetic reaction. In all experiments, glucosamine was used in its salt form, requiring deprotonation for nucleophilic activation. Reaction 4, employing pyridine as a base and a small volume of DMF at 65 °C, led to the most promising results. The addition of DMF was necessary to improve the dissolution of the substrates, which was also favored by an increase in temperature.

Figure 10 reports a comparison of the ^1^H NMR spectra in DMSO obtained from the starting PGAI polymer, reference glucosamine, and the reaction product. Signals pertaining to glucosamine (indicated by black dots in the figure) are present in the NMR spectrum of the reaction product, suggesting the successful coupling of GA with PGAI. This is further confirmed by an analysis of the PGSTE spectra of PGAI and of PGAI-GA obtained with the highest applied gradient (Appendix A, ESI). The permanence of the glucosamine signals in the PGSTE spectrum of the product further points towards the coupling of GA and PGAI. Observing the integrals of the signals of the vinyl protons of dimethyl itaconate (Figure 11), a 17% decrease in the intensity of the signal is observed after reaction with glucosamine, further pointing towards the formation of a bond between the DMI and the amino group of glucosamine.

The choice of the base was pivotal: triethylamine, a stronger, less hindered base, immediately and irreversibly induced isomerization of the vinyl group to internal double bond structures (namely, mesaconate and citraconate derivatives), which are significantly less reactive in nucleophilic addition reactions. This effect was evidenced by 1H-NMR and confirmed the base-induced rearrangements of itaconic acid [35,36].

## 4. Conclusions

Post-polymerization functionalization of bio-based polyesters and oligomers offers a wide range of options for designing and producing new, advanced macromolecules and materials for use in controlled drug release, targeted delivery, tissue engineering, cosmetic formulations, smart coatings, and packaging. This study presents a multidisciplinary, integrated strategy for the rational ecodesign and synthesis of bio-based, unsaturated terpolymers functionalized with various biomolecules. This study aimed to demonstrate how the design and production of a new chemical product can be planned with the ‘safety and sustainability’ of the chemicals in mind from the initial stages of the implementation process.

The proposed approach starts with the following: (i) in silico selection of the most promising candidates for a given application by virtual screening of candidates oligoesters using Volsurf+ software, which can predict physical and chemical properties, such as hydrophilicity, flexibility, permeability; (ii) solvent-free enzymatic synthesis of the selected bio-based oligoester under mild (50 °C) and solvent-free conditions to obtain short terpolymers (between 400 and 1921 Da) with a controlled structure (polydispersity close to 1) containing reactive monomers (in our case the vinyl group of itaconic acid), which can be exploited as a chemical platform for further functionalization; (iii) study of the marine biodegradability and ecotoxicity of the oligoesters towards freshwater and marine benchmark organisms, given that biodegradability does not necessarily indicate low toxicity, and the fate of most pharmaceutical and cosmetic products occurs in these ecosystems; and (iv) covalent anchoring of biomolecules on the unsaturated terpolymer via Michael addition reactions.

Computational predictions were useful for the selection of poly(glycerol adipate itaconate) as the most suitable candidate due to its higher predicted hydrophilicity and flexibility. These properties were supported by experiments that successfully synthesized PGAI under mild conditions. This synthesis preserved the vinyl functionality, which enables post-polymerization functionalization and covalently anchors biomolecules via Michael addition. Furthermore, PGAI’s biodegradability and negligible ecotoxicity confirmed that the computationally pre-screened candidate translates into a safe and sustainable chemical platform.

The multidisciplinary planning of the research allowed the experimental efforts to focus on targets that are promising not only in terms of technical performance but also safe and sustainable. Therefore, the in-depth simulation of the products by DFT and MD simulations is out of the objective of the present paper, although this will be the aim of our future work, which will be focused on the selected molecules. This shows that, when coupled with experimental validation, descriptor-based predictions can reliably support the rational ecodesign of bio-based oligoesters and reduce the consumption of resources.

The experimental results confirm that lipase-catalyzed polycondensation is an effective method of introducing itaconic units into polyesters while preserving the vinyl group. It is also worth noting that previous studies have demonstrated the recyclability of lipase CaLB when it is covalently immobilized on fossil-based and bio-based carriers, making the process economical [12,15,23,24]. Overall, this study contributes to the understanding of the reactivity of different nucleophilic groups of selected bioactive molecules in the thia- and aza-Michael reactions. Although the Michael reactions were not optimized, NMR data indicate that the *exo* C=C fully reacted when N-acetylcysteine and N-Ac-Phe-ε-Lys-OtBu were used as nucleophiles. In the latter case, the hindered α-amino group of the dipeptide was covalently anchored on the terpolymer, opening new avenues for the functionalization of polyesters with amino acids and peptides. In the case of glucosamine, the primary amine reacted, albeit with lower conversions (approximately 19%), due to the substrate’s low solubility and mass transfer limitations. Finally, results indicate that the addition of trilysine to PGAI is not viable because of the presence of three unhindered amino groups that promote the formation of secondary cyclic products and cross-links. These preliminary data suggest that there is room for further investigations and optimization towards the effective binding of bioactive molecules on polyesters containing itaconic units, provided that the nucleophiles are either the thiol group or hindered primary amines.

The multidisciplinary strategy here presented can be adapted to suit a wide range of biological or technological objectives. The combination of mild enzymatic polycondensation conditions with the low ecotoxicity and high biodegradability of various bio-based oligoesters [5] shows promise in the development of different ‘safe and sustainable’ chemical platforms for advanced products and materials.

## Data Availability

Raw data of the computational analysis and of the ecotoxicity studies are available in Appendix A.

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
