# Peer review of "Integrating Computational and Experimental Methods for the Rational Ecodesign and Synthesis of Functionalized Safe and Sustainable Biobased Oligoesters"

_polymers, 2025, doi:10.3390/polym17182537_

Round 1

Reviewer 1 Report

Comments and Suggestions for Authors

This manuscript primarily utilizes VolSurf+ software to select oligoesters, evaluating hydrophobicity, flexibility, and permeability. The AA-GLY-IA ternary oligoester is proposed as optimal. However, beyond virtual data and experimental NMR/ MS characterisation, the manuscript lacks direct experimental evidence supporting the three key properties – hydrophobicity, flexibility, and permeability. For instance: contact angle measurements on thin-film samples could verify hydrophobicity, and DMA (Dynamic Mechanical Analysis) could demonstrate flexibility.

Furthermore, Figure 3 lacks units on its axes. This means the presented data can only indicate relative differences in predicted values across these systems examined, not absolute experimental data.
Considering the authors' potentially limited experimental capabilities, if the study relies predominantly on virtual experiments, it is strongly recommended to supplement the work with DFT (Density Functional Theory) and MD (Molecular Dynamics) simulations on selected representative systems in Figure 3 to provide computational evidence for the hydrophobicity, flexibility, and permeability properties. Otherwise, the manuscript in its current form could not be published in Polymers.

Additionally, the manuscript contains several minor issues as detailed below:

  1. The reference citation should be corrected instead of displaying “Error! Reference source not found...”, such as on page 7, line 268; page 11, line 372, etc.
  2. In Supplementary Tables S1 and S2, column headers lack explicit definitions – notably V (Column 2), S (Column 3), W1 (Column 5).
  3. Regarding Supplementary Figures S1, S2, etc., the authors state that the NMR spectra were acquired using DMSO-d6 as the solvent. However, the characteristic solvent peak of DMSO at 2.50 ppm and the water peak at 3.33 ppm are absent. We recommend that the authors carefully verify all NMR results.

Reviewer 2 Report

Comments and Suggestions for Authors

The paper "Integrating computational and experimental methods for the rational ecodesign and synthesis of functionalized safe and sustainable biobased oligoesters" reports the design,preparation and functionalization of new oligoesters based on poly(glycerol adipate itaconate) via enzymatic and aza/thia-Michael addition, respectively.

I appreciate the overall work reported here and the characterization methods to demonstrate the structure of the products taking into account the affinity of the starting molecules by computational analysis.

I identified only a few minor issues:

Abstract: a series of abbreviations must be defined: ADME, N-Ac-Phe-ε-Lys-OtBu, Lys-Lys-Lys.

Introduction: is too general, there are some information related to the own works. The limitation of these studies must be mentioned, the advantages of the proposed methods must be highlighted!

Section 3. The data shown in Figure 3 must be tabulated.

There are a series of errors in the main text at lines: 372-373; 381; 532;

Section 3.2. The yield of the product must be mentioned!

Figure 4: the resolution of the text is low, I suggest a table including the main m/z values of the product highlighted in red.

Figure 5. The resolution is low and the letters are not visible.

The yields of the functionalised products must be mentioned!

The resolution of the 1HNMR spectra must be improved!

A mechanism of biodegradation must be proposed!

Conclusions. To what content the predicted data have been demonstrated by the synthetic products?

Based on these considerations my recommendation is Minor revision!

Round 2

Reviewer 1 Report

Comments and Suggestions for Authors

Now it can be published. Congulation!